# The Effect of Walnut Consumption on *n*-3 Fatty Acid Profile of Healthy People Living in a Non-Mediterranean West Balkan Country, a Small Scale Randomized Study

**DOI:** 10.3390/nu12010192

**Published:** 2020-01-10

**Authors:** Gordana Petrović-Oggiano, Jasmina Debeljak-Martačić, Slavica Ranković, Biljana Pokimica, Alma Mirić, Maria Glibetić, Tamara Popović

**Affiliations:** Center of Research Excellence in Nutrition and Metabolism, Institute for Medical Research, University of Belgrade, 11000 Belgrade, Serbia; minaizdravko@yahoo.com (J.D.-M.); slavica.rankovic.imr@gmail.com (S.R.); biljana.pokimica@hotmail.com (B.P.); savrshenstvovatre@yahoo.com (A.M.); mglibetic@gmail.com (M.G.); poptam@gmail.com (T.P.)

**Keywords:** *n*-3 PUFA, walnut consumption, non-Mediterranean West Balkan country, small scale randomized study

## Abstract

People living in non-Mediterranean West Balkan countries have diets with a low *n*-3 polyunsaturated fatty acid (PUFA) content. Walnuts, a traditional Serbian food, could be an excellent source of *n*-3 PUFA. The first sub-study evaluated the fatty acid and mineral content of Serbian walnuts, demonstrating that walnuts had the high content of linolenic acid (C18:3, *n*-3 ALA). The second sub-study assessed the consumption of walnuts (*Juglans regia* L.) and total *n*-3-fatty acid intake in apparently healthy Serbian residents, using 24-h dietary recalls (*n* = 352). An inadequate intake of *n*-3 fatty acids and a low consumption of walnuts was seen. Additionally, we evaluated the fatty acid profile of healthy Serbian adults (*n* = 110) and finally, via a randomized intervention 4-weeks study, we assessed the effects of walnut consumption on *n*-3 fatty acid profile of participants (*n* = 18). The plasma content of *n*-3 PUFA was low and the *n*-6/*n*-3 ratio was high in our study participants. The *n*-3 plasma fatty acid profile was improved after 4 weeks of walnut consumption, meaning that ALA, eicosapentaenoic acid, and total *n*-3 were significantly increased. The results of our study pointed out the potential health benefits of walnuts consumption on amelioration of the *n*-3 fatty acid profile that should be taken into account in preventive management programs. The higher conversion of ALA to EPA (>10%) in examined study participants, suggests the importance of a moderate walnut consumption.

## 1. Introduction

Walnuts (*Juglans regia* L.) are grown in Serbia and are a traditional Serbian food [1]. Walnuts are a nutrient-dense food containing many bioactive constituents which separately or together may produce beneficial effects on human health [2]. Based on the analysis of the EuroFir food composition databases, walnuts were identified as major sources of *n*-3 fatty acids among nuts in the Balkan region. Additionally, walnuts represent nuts rich in ALA. Maguire and coauthors, showed that walnuts have particularly high content of ALA (11.6% of total) compared to other nuts (i.e. hazelnuts and almonds) [3,4]. The ratio of ALA to linoleic acid is high among all the tree nuts [5] meaning that walnuts have a beneficial *n*-6/*n*-3 ratio [6]. Given the cultural acceptability of walnuts as a traditional food, walnuts are seen as an excellent candidate for improving *n*-3 fatty acid intake [7]. Limited consumption of *n*-3 fatty acids and a Western type diet contribute to the unfavorable fatty acid profile in humans. The intake of *n*-6 PUFA is significantly higher than the intake of *n*-3 PUFA which results in a negative shift of *n*-6/*n*-3 ratio in plasma [1]. In addition to *n*-3 fatty acids, walnuts contain many different minerals [8,9]. The walnuts are a good source of manganese, a trace element important for maintaining cellular redox status [10]. By virtue of the unique micro- and macro-nutrient composition of walnuts, an increased consumption of walnuts has been seen as beneficial for cardiovascular health [11]. Prospective observational studies and large clinical trials demonstrate various health benefits of walnut consumption including a reduced risk of cardiovascular diseases, cholesterol-lowering effects, reduced oxidative stress, reduced inflammation, and blood pressure and an increased vascular reactivity [12]. In order to overcome the low intake of *n*-3 FAs walnuts could be a beneficial dietary choice. Therefore, we postulated the following aims: The first aim was to analyze the nutritional composition data, fatty acid and mineral content of walnuts grown in Serbia. The second aim was to evaluate the consumption of *n*-3-fatty acids by Serbian residents using the 24-h dietary recalls. The third aim was to evaluate the fatty acid profile of an apparently healthy group of people. The fourth aim was to assess the effect of walnut consumption in the amount of 56 g daily on the plasma phospholipid fatty acid profile of participants involved in a randomized 4-weeks trial. 

## 2. Materials and Methods 

### 2.1. Walnut Samples

The Sampling plan was performed according to the EuroFIR defined criteria. Walnuts (*Juglans regia* L.) were collected from the four representative markets in ten biggest cities in Serbia. The primary sample was generated by mixing an equal portion of four samples taken from those markets. A composite sample was defined as a mix of equal portion (100 g) of 10 primary samples.

Five replicate samples of the composite samples were analyzed further for nutrient and bioactive content in ground walnuts [13]. All values are presented as mean ± SD. 

#### 2.1.1. Determination of Fatty Acids in Walnuts Total Lipids

Total lipids were extracted according to method of Folch [14] using 2:1 chloroform:methanol mixture containing butylated hydroxytoluene (0.05% BHT weight/volume).

Fatty acid methyl esters of fatty acids were prepared by transmethylation with sodium hydroxide (2 M) in methanol (heated at 85 °C for 1 h) followed by heating in sulfuric acid (1 M) in methanol (heated 85 °C for 2 h). After 30 min, samples were centrifuged and upper phase of samples were put into tubes and evaporated with technical nitrogen. Fatty acid methyl esters were extracted into hexane before analysis. Separations of the methyl esters were carried out using a gas chromatograph (Shimadzu, Kyoto, Japan), equipped with a split/splitless injector and a flame ionization detector. The methyl ester separation was carried out on capillary column RTX 2330 column (60 m × 0.25 mm with a 0.20 μm film) from RESTEK, Bellefonte, PA using helium as the carrier gas. The injector and detector temperature was set at 220 °C and 260 °C, respectively. The injection was performed in a split mode with a 1:20 split-ratio. The temperature of the column was initially set at 140 °C for 5 min, and then increased to 220 °C at the rate of 3 °C /min, and held at this temperature for 20 min. Each of the fatty acids was identified with a reference to the retention time of that in a PUFA-2 standard mixture (Sigma-Aldrich, St. Louis, MO, USA). The content of individual FAs was expressed as a percentage of the total fatty acids as previously described [1].

#### 2.1.2. Determination of Macronutrients in Walnuts

The nutritional analysis was carried out by an accredited chemical laboratory at the Institute of Public Health in Požarevac. The ash content was determined by a direct gravimetric method (AOAC, 2000, Rockville, MD, USA) [15] (AOAC 999.11) [16] that includes ashing of the samples in an oven at 550 °C until the constant weight was attained. The moisture content was determined gravimetrically [15,16]. The crude protein content was estimated based on the total nitrogen content of samples determined by Kjeldahl method [17]. Crude fat content was determined gravimetrically (Soxhlet extraction, AOAC method (AOAC 963.15) [18]. Total carbohydrate content, crude “by difference”, was calculated by the following formula: Total carbohydrate (%) = 100% − % (protein + ash + fat + moisture). The energy content of walnuts was calculated based on determined content by the following formula: Energy value (estimated, kJ/100 g) = [4 × protein (%)] + [4 × carbohydrate (%)] + [9 × fat (%)]. 

#### 2.1.3. Mineral Composition of Walnuts

The content of minerals in the samples was quantified with atomic absorption spectrometry (Spectra AA 10, Varian Pty Ltd., Mulgrave Victoria, Australia). Before the spectrometric analyses, samples were ground and homogenized. Approximately 5 g of each dry sample (in duplicate) was weighed and mineralized by dry ashing on 450 °C in a furnace (PF 1, Vecstar LTD, Chesterfield, UK) according to the AOAC official method of analysis [19] (AOAC 999.11, 2000). The mineral contents of all mineralized samples were determined in duplicates on atomic absorption spectrometer (Spectra AA 10, Varian Pty Ltd., Mulgrave Victoria, Australia) in an air acetylene flame (Ca, Mg, Na, K, Cu, Fe, Mn, Zn, Cr, Ni) using deuterium background correction. For the determination of Na and K, cesium was added to standards and samples as an ionization buffer (0.2% w/v). For Ca and Mg measurements, La was used as a releasing agent (0.1% w/v). Measurements were performed at the following wavelengths (nm): Ca 422.7, Mg 285.2, Na 589.0, K 766.5, Cu 324.7, Fe 248.3, Mn 279.5, Zn 213.9, Ni 232.0, and Cr 357.9. Reagent blanks and samples spiked with element standards were routinely included in the analysis.

### 2.2. Assessment of Dietary Intake of Walnuts and n-3 Fatty Acids 

Repeated 24 h dietary recalls obtained for two non-consecutive days were used to assess the consumption of walnuts among the study participants (Figure 1). In addition, the intake of total *n*-3-fatty acids was evaluated. Data were analyzed by the Diet Assess and Plan software using the Serbian Food Composition Data Base [19,20], in compliance with EU methodology [21]. SPADE software was used to calculate usual/habitual dietary intake of *n*-3 FA [22].

### 2.3. Study Participants and Collection of Blood Samples.

The study participants were apparently healthy volunteers recruited during January–February 2017, through primary health care facilities where flyers about participation in the study were distributed. ‘Apparently healthy’ was defined as the absence of any clinical signs of acute condition or chronic disease and without the need for a medical treatment. A total of 400 subjects were approached for the recruitment via the Institute for medical research and 352 were involved in the dietary data analysis (Figure 1). The study sample consisted of 75 children aged from 10 to 17 years; 235 adults aged 18 to 64 years and 42 seniors aged 65 and older. All of the study participants have signed the informed consent forms. The study protocol was approved by the Institute for medical research, Belgrade, Serbia, Ethics Committee Approval No: 0173/17). The study was conducted in agreement with the ethical guidelines on biomedical research on human subjects of The Code of Ethics of the World Medical Association’s Declaration of Helsinki and its further amendments. 

For the analysis of fatty acids in plasma phospholipids 30% of initially enrolled subjects were randomly selected. Blood samples were obtained from 114 participants. 

At the later stage, eighteen participants (15% of 110) were involved in a 4-weeks randomized intervention study, during which they consumed 56 g of walnuts daily (Part of registration study NCT03227497). As our intervention study was a small scale (pilot study) 10–15% of study participants was intended to be included (11–18 participants). We have to mention here that due to restricted resources we were not able to include more participants within this interventional sub-study. The amount of 56 g was chosen based on the literature data [11,12,23]. Participants were instructed to consume their usual diet, with an exclusion of other nuts (i.e., almonds, hazelnuts) from their diet and asked to bring empty plastic bags in order to check compliance. Blood samples were collected after an overnight fast, blood was drawn into EDTA tubes and plasma samples were stored at −80 °C until further analysis. 

#### 2.3.1. Anthropometric and Blood Pressure Measurements 

Anthropometric measurements were performed using standard procedures [24], including body height, body weight, and waist circumference (WC). Body weight was determined using a bioelectrical impedance analyzer, TANITA UM072 balance (TANITA Health Equipment Hong Kong, China). Participants were asked to remove outer clothing and shoes and instructed to stand in the center of the analyzer’s platform so they could weight would be evenly distributed on both feet. Height was measured using a height measuring scale. The WC was measured at the umbilicus level, allowing the measurement tape to be horizontal to the floor. The cut-off value for elevated WC was ≥94 cm for men and ≥80 cm for women [24]; for elevated WHR it was ≥0.85 for women and ≥0.90 for men [25]. Systolic blood pressure (SBP) and diastolic blood pressure (DBP) were determined using a digital upper arm electronic blood pressure monitor (OMRON, HEM-907, Omron Healthcare, Tokyo, Japan) in resting subjects in the sitting position. Blood pressure values were obtained in triplicates separated by a 2 min break and the average values of the three measurements was recorded. All measurements were taken at the baseline, and at the end of the study. Study participants were considered to have hypertension if they (a) reported a previous diagnosis of hypertension or (b) reported the use of anti-hypertensive medication or (c) had a mean systolic blood pressure ≥140 mmHg or mean diastolic pressure ≥90 mmHg [26]. 

#### 2.3.2. Determination of Fatty Acids in Plasma Phospholipids of Study Subjects

Total lipids were extracted according to method of Folch [14] using 2:1 chloroform: methanol mixture containing butylatedhydroxytoluene (0.05% BHT weight/volume). The phospholipids fraction from plasma was isolated from extracted lipids by one-dimensional TLC with neutral lipid solvent system of hexane:diethylether:acetic acid (87:2:1) using Silica Gel GF plates (C. Merck, Darmstadt, Germany). Fatty acids analyzed as per Section 2.1.1. The content of FA was expressed as percentage of total fatty acids as previously described [1].

### 2.4. Statistical Analysis 

The data were tested for normality by Shapiro–Wilk Test. Wilkoxon signed-rank test was used for non-normally distributed data All analysis were performed in SPSS version 23.0 (Chicago, IL, USA) and a 2-tailed *p* value of <0.05 was considered statistically significant.

## 3. Results 

### 3.1. Content of Macronutrients, Minerals, and Fatty Acids in Walnuts

The mean macronutrient composition of walnuts is shown in Table 1. Presented data are similar to the results reported in the European Food Information Resource (EuroFIR databases, 2013), demonstrating a high content of fat in walnuts. 

The results showed (Table 2) that walnut cultivars from Serbia were similar to the walnut cultivars in other EuroFIR- countries in terms of their fatty acid content, and linolenic (C18:3, *n*-3) to linoleic (C18:2, *n*-6) fatty acid ratio (https://www.eurofir.org/).

The evaluation of the levels of essentials minerals in walnuts from markets in Serbia, is shown in Table 3. 

### 3.2. Consumption of Walnuts and Total n-3 Fatty Acid Intake of Study Participants 

Only 20% of the study participants consumed whole walnuts. Among our study participants, walnuts consumers had an intake of 9.7 g/d, on average. At the level of the whole study sample, the intake was significantly lower, 1.5 g/d. The intake of *n*-3 FAs in our study group was 0.9 g/d.

### 3.3. Analysis of Plasma Phospholipid Fatty Acid Composition 

According to our analysis (Table 4) the low *n*-3 PUFA followed by the high *n*-6 PUFA intake resulted in the high *n*-6/*n*-3 ratio in plasma phospholipids of our study participants. 

### 3.4. The Effects of Walnut Consumption on Anthropometric and Biochemical Parameters and Plasma Phospholipid Fatty Acids 

The present study showed statistically significant decreases in BMI, percentage of body fat and SBP (Table 5). Results regarding the fatty acid composition (Table 6) revealed an increased percentages of ALA (*p* ≤ 0.01), EPA (*p* ≤ 0.01) and overall *n*-3 (*p* ≤ 0.05) in plasma phospholipids, while the *n*-6/*n*-3 ratio was decreased (*p* ≤ 0.05).

## 4. Discussion

This study confirmed that walnuts are a good source of *n*-3 PUFA. People living in Serbia have a tendency to have a low dietary intake of *n*-3 PUFA which was confirmed by FA profile analysis. Consumption of walnuts short term (4 weeks) could be beneficial for increasing the intake of *n*-3 PUFA. 

Walnuts contain many nutrients that could be helpful for improving fatty acid, phytochemical and micronutrient intake of consumers [27]. The variability between walnuts obtained from different markets is larger than for fatty acids, resulting in a higher coefficient of variation. The results obtained in the present study are similar to data presented within the European Food Information and resource database. With this study we added nutrient composition data for manganese, nickel and sodium. In comparison of the mineral content of Serbian walnuts with that of other European countries, using the simultaneous access to a multiple set of available Food composition data base (FCDBs) through the EuroFIR FoodEXplorer (2013) [28], it has been shown that there are some differences, also similarity, which affect nutrition composition database.

Although the major fatty acid in walnuts is linoleic acid (*n*-6), the content of linolenic acid (*n*-3), is also substantial [6]. Walnuts contain higher amounts of linolenic acid (*n*-3) than any other nuts (almonds, hazelnuts) [29]. According to the values presented in Table 2, a 25-gram serving of Serbian walnuts would provide 1.9 g of linolenic acid which is an amount well within the recommended intake of 0.58 to 2.81 g/day [30]. Data presented by Marangoni et al., [31] indicate that the intake of slightly over 1 g ALA/day through walnut consumption significantly and rapidly increased blood ALA and EPA levels. 

We evaluated the intake of foods rich in *n*-3 fatty acids, precisely the intake of walnuts. Only 20% of the study participants consumed whole walnuts. Among our study participants, walnuts consumers ate on average 9.7 g/d. On the level of the whole study sample, the intake is significantly lower to 1.5 g/d. Among the US adult population, an average consumption of 10.3 g/d and 13.1 g/d of walnuts was reported for walnut consumers 29–59 years old and those older than 60 years, respectively (Arab, Ang, 2015). Authors of PREDIMED study, reported that the average intake of walnuts per day for all study participants was 5.9 g [32]. Probably, the majority of the population consume walnuts incorporated in some other foods, such as cakes.

The mean dietary ratio of *n*-6 to *n*-3 fatty acids in a sample of Serbian individuals was high (21.3), confirming poor eating patterns of people in this region. Some evidence suggest that *n*-6/*n*-3 ratio may be a potential biomarker for the development of cardiovascular diseases and decreasing the ratio could improve health status of Serbian population [1]. There are published data on the *n*-6/*n*-3 ratio in some other countries like Japan (4.00), United Kingdom and Northern Europe (15.00), the United States (16.74), and urban India (38–50) [33]. In general people living in Serbia, a non-Mediterranean West Balkan country, have an inadequate intake of total *n*-3 fatty acids [34]. Data from the European Prospective Investigations into Cancer and Nutrition (EPIC)—Norfolk Study showed that the total dietary intake of *n*-3 FAs in men was 1.5 g/d, while in women it was 1.22 g/d [35]. PREDIMED study pointed out that in Spain, the average daily dietary *n*-3 intake was 2.24 g (calculated as a sum of ALA and *n*-3 LC PUFA [32]. The intake of *n*-3 FAs in our study group was 0.9 g/d, which is substantially lower from the intakes reported for aforementioned Mediterranean countries. The ability of people to meet the recommended daily intake of *n*-3 fatty acids via consumption of walnuts is important in non-Mediterranean West Balkan countries where a Greek-Mediterranean diet is not typically consumed. As the incidence of cardiovascular diseases in Serbia is very high [36,37], an increased consumption of *n*-3 fatty acid rich foods, such as walnuts, could be of great importance. Potential interventions aimed at increasing the consumption of walnuts might be more easily implemented in Serbia than in other countries, since the walnuts are a part of traditional Serbian food [1].

Although the major fatty acid in walnuts is linoleic acid (*n*-6), the content of linolenic acid (*n*-3), is substantial [6]. Walnuts contain higher amounts of linolenic acid (*n*-3) than any other nuts (almonds, hazelnuts, chestnuts) [38]. Using the values in Table 2, a 25-gram serving of Serbian walnuts would provide 1.9 g of linolenic acid which is an amount well within the recommended intake of 0.58 to 2.81 g/day [39]. Data presented by Marangoni et al., [31] indicate that the intake of slightly over 1 g ALA/day through walnut significantly and rapidly increased blood ALA and EPA levels. The efficiency of the pathway is inherently low in humans, with an estimated conversion of ALA to EPA of 0.2–6% and <0.1% for DHA, and therefore any changes in bioconversion efficiency have potentially large impacts on LC PUFA status [40]. We hypothesized that ALA from walnuts could correct the low *n*-3 status and tested this by a short-term intervention pilot study. Our pilot study data showed the improvement in the percentages of ALA (*p* ≤ 0.01), EPA (*p* ≤ 0.01) and overall *n*-3 (*p* ≤ 0.05) FAs in plasma phospholipids of study participants, while the *n*-6/*n*-3 ratio was decreased (*p* ≤ 0.05).

Regarding the impact of consumption of walnuts on anthropometric measures our study demonstrated (Table 5), a significant reduction in BMI, percentage of body fat, increased lean body mass and an increased amount of water in the body. Similar results (i.e., weight reduction) was provided by a large population-based cohort study (−0.10 kg, 95% CI-0.15, −0.04) [23,41]. Regarding the impact of walnut consumption on blood pressure measurements, we detected a significant reduction in systolic blood pressure (Table 5). The blood pressure lowering effect could be due to the increased content of minerals in walnuts, i.e., magnesium and potassium. The impact of walnuts nutritional intervention on decreasing effect of blood pressure is also observed in the previous intervention the “Health Track study”, which reported significant reductions of SBP in examined study participants, which was attributed to the intake of 30 g walnuts/d [42]. 

Our study results suggest that the moderate consumption, of walnuts not only improves the *n*-3 fatty acid status of consumers, people of West Balkan non Mediterranean countries, but could also have some additional beneficial effects on human health. It is important to point out that there was an increase in EPA indicating an elevated conversion of ALA to EPA (>10%), Table 5.

Regarding the study limitations, we underline a short-term duration of our intervention study and a small number of the participants included in this sub-study, so in this context the data should be interpreted with caution. The higher conversion of ALA to EPA (>10%) in examined study participants underlines the importance of a moderate consumption of walnuts. In addition, the obtained data provide basis for further larger clinical trials. Future studies should be designed with a higher number of study participants and should be run over a longer period of time. 

## 5. Conclusions

We have confirmed that walnuts (*Juglans regia* L.) are a good source of *n*-3 PUFA, especially ALA. Regarding the mineral content of the walnuts, they are an especially good source of magnesium and manganese. This will impact on the correct calculation of nutrient intakes in clinical and population-based studies. It will also impact on the use of walnuts in dietetic therapy to meet the recommended nutrient intakes. Also, apparently healthy Serbian residents have an inadequate intake of *n*-3 fatty acids, that is also reflected in plasma phospholipid fatty acid profile. The results of our short-term pilot study indicate possible beneficial associations between the moderate intake of walnuts (56 g/per day) and the improvement of *n*-3 fatty acid profile, especially favorable impact of ALA and EPA in the plasma phospholipids. In the future, in order to elucidate the actual mechanisms, a larger size epidemiological and/or well controlled experimental studies of longer duration are needed. 

### Study Limitations

Regarding the study limitations, we underline short-term period and small number of the participants which did not have a control group and the results should be interpreted with caution.

Additionally, regarding the duration of the intervention, most published studies pointed out the effect of walnuts on fatty acid profiles during a period of few months [43] and therefore, cause more substantial and relevant changes. Future studies should be designed with higher number of study participants and longer intervention periods. The important novelty was a higher conversion in ALA to EPA (>10%) in examined study participants, which underlines the importance of moderate nut consumption.

## Figures and Tables

**Figure 1 nutrients-12-00192-f001:**
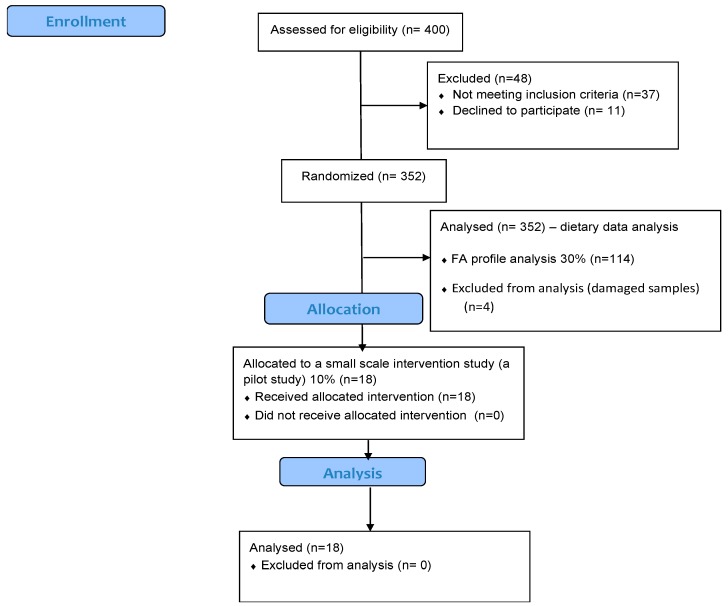
Flow chart indicating the number of participants.

**Table 1 nutrients-12-00192-t001:** Mean macronutrient content of Serbian walnuts.

Macronutrient	Mean (SD)
Carbohydrate	8.10 (0.22)
Protein	16.21 (0.05)
Fat	68.38 (0.04)
Water	3.60 (0.02)
Ash	1.84 (0.01)
Fiber	1.86 (0.11)

Values are expressed as % by weight. Abbreviations: SD, standard deviation.

**Table 2 nutrients-12-00192-t002:** Fatty acid composition of composite samples from different markets in Serbia.

Fatty Acid	16:0	16:1	18:0	18:1, *n*-9	18:1, *n*-7	18:2, *n*-6	18:3, *n*-3
Serbia	7.03 (0.25)	0.11 (0.05)	2.75 (0.26)	14.47 (1.17)	1.34 (0.39)	63.15 (0.93)	11.15 (0.71)

Values obtained were expressed as % of total fatty acids. Values were represented as means (SD). Abbreviations 16:0, palmitic acid; 16:1, palmitoleic acid; 18:0, stearic acid; 18:1, *n*-9, oleic acid; 18:1, *n*-7, vaccenic acid; 18:2,*n*-6 linoleic acid; 18:3, *n*-3, alpha-linolenic acid; n.a., not available; SD, standard deviation.

**Table 3 nutrients-12-00192-t003:** Mineral composition of walnuts from Serbia.

Mineral	Cu (mg/100 g)	Mn (mg/100 g)	Fe (mg/100 g)	Zn (mg/100 g)	Ni (mg/100 g)	Cr (µg/100 g)	Na (mg/100 g)	K (mg/100 g)	Ca (mg/100 g)	Mg (mg/100 g)
Mean (SD)	1.29 (0.29)	3.45 (1.28)	2.20 (0.11)	4.52 (3.63)	0.29 (0.16)	0.022 (0.02)	14.2 (1.90)	438.2 (60.90)	113.5 (42.30)	147.2 (15.50)

Abbreviations: Ca, calcium; Cu, copper; Cr, chromium; Fe, iron; K, potassium.

**Table 4 nutrients-12-00192-t004:** Fatty acid composition of plasma phospholipids in a sample of Serbian adults (*n* = 110), 68 females, mean age 48 years, mean BMI 29.6 kg/m^2^ (females), and 28.5 kg/m^2^ (males).

Fatty Acids (%)	Mean (SD)
SFA	46.9 (5.00)
MUFA	10.8 (1.30)
PUFA	41.8 (2.90)
*n*-3	3.6 (1.10)
*n*-6	38.3 (2.80)
*n*-6/*n*-3	11.6 (3.60)

Abbreviations: MUFA, monounsaturated fatty acids; PUFA, polyunsaturated fatty acids; SD, standard deviation; SFA, saturated fatty acids.

**Table 5 nutrients-12-00192-t005:** Changes in parameters of body composition and blood pressure (at the baseline and after the dietary intervention) in study participants.

	Baseline	End of Intervention	*p*-Value
Sex; *n* = 18			
Men	9	9	
Women	9	9	
Age (years)	47.1 ± 5.06		
BMI (kg/m^2^)	27.73 ± 3.5	27.14 ± 3.49	0.0064 **
WC (cm)	91.75 ± 13.62	89.50 ± 13	0.1473 ns
FAT MASS (kg)	32.98 ± 8.54	31.36 ± 8.51	0.0006 ***
Lean mass (FFM) (kg)	48.15 ± 17.55	49.80 ± 17.72	0.0033 **
% Water	48.48 ± 5.31	49.63 ± 5.39	0.0008 ***
SBP (mmHg)	137.50 ± 17.00	128.3 ± 13.74	0.0015 **
DBP (mmHg)	76.00 ± 20	74.00 ± 19.25	0.0793 ns

Data are presented as mean ± SD if normally distributed, otherwise as median (IQR). *p* value ** (*p* ≤ 0.01), *** (*p* ≤ 0.001).

**Table 6 nutrients-12-00192-t006:** Fatty acids in plasma phospholipids of study participants before and after the intervention.

Fatty Acids (%)	Before Treatment MEDIAN (IQR)	After Treatment MEDIAN (IQR)	*p* Value
ALA	0.107 (0.064–0.142)	0.163 (0.130–0.190)	0.005 **
EPA	0.342 (0.230–0.450)	0.440 (0.300–0.590)	0.010
DHA	2.494 (2.001–3.777)	2.811 (2.467–3.978)	0.094
*n*-3	3.426 (3.054–5.207)	3.895(3.621–5.391)	0.029
*n*-6	38.723 (37.391–41.644)	39.221 (37.262–40.491)	0.601
*n*-6/*n*-3	11.53 (7.503–12.510)	9.534 (7.511–11.080)	0.021

Abbreviations: ALA-alpha linolenic acid, EPA-eicosapentaenoic acid, DHA-docosahexaenoic acid, IQR-Interquartile range, *p* value ** (*p* < 001).

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
