# Peer review of "The Effect of Walnut Consumption on n-3 Fatty Acid Profile of Healthy People Living in a Non-Mediterranean West Balkan Country, a Small Scale Randomized Study"

_nutrients, 2020, doi:10.3390/nu12010192_

Round 1
Reviewer 1 Report
I have reviewed the manuscript "Improvement of n-3 status by increased walnut consumption in countries with low fish intake".
While the data seems to be relevant and of interest for the readership of Nutrients, the overall quality of the manuscript itself is very low in its present form. Writing and presentation of data is unfriendly to a broad readership, and it takes serious efforts to find the most relevant information for an appropriate scientific report, in order to allow adequate evaluation whether it is indeed possible to draw the conclusions proposed by the authors.
The manuscript needs a rigorous revision providing an introduction that clearly states the background knowledge on the topic (concise, using appropriate references), the gap in the field (clearly stated in 1-2 sentences), the hypothesis to be studied, and the aims of the study (which the authors have rather provided as a blurry mixed between objectives, aims, methods, etc). Please be very clear with the aforementioned, particularly with the aims of the study because this manuscript overall feels like a mix of attempts to do many things (some of them actually seem irrelevant and rather obscure the primary intetion of the study, but it is not clear whether this is due to the writing or due to scientific issues). Do not include in this section information that would rather belong to the discussion section.
The methods section needs to start with clear description of the study population, and all the relevant information regarding recruitment, including the settings in which patients were approached for participation (please realize that mentioning the name of the institution is poor information for an international audience, it does not inform the reader more than the location of the study), how many patients were approached, how many rejected participation, etc. A flowchart is needed. Remarkably, it is not clear how the authors arrived to the number of patients that provided blood samples (110 out of the 352 initial participants), or those that participated in the very much unclear interventional step of the study (18 out of 352 initial participants).
I have serious concerns regarding the intervention because if I understand correctly, patients were asked to not consume fish during the period of the intervention. Considering that fish intake is one important element for a healthy diet as recommended in several international guidelines, my concern is regarding the ethical approval of doing this. This needs to be clarified and appropriately supported.
Also regarding this, the correct term is "informed consent", not "confirmed consent". It is also not appropriate to use the term "treatment" when referring to the "intervention".
The authors should use an international guideline (e.g. an applicable STROBE checklist) to ensure a better quality of reporting.
Finally, please downturn and be very careful with the conclusions. In its current writing, the last paragraph of the manuscript, under the heading conclusions, clearly overstates what would be at best possible to conclude from the performed study, and this needs to be rigorously revised before any further consideration of the manuscript.
Last but not least, the title of the manuscript also overstates the study and seriously misleading because rather than studying "countries with low fish intake", the authors only studied a relatively small sample population of a single-center in only one country.
Author Response
Answers to Reviewer 1
Dear Reviewer,
We are grateful for your usuful and very constructive suggestions for our article named:
Effect of walnuts consumption on improvement of n-3 fatty acid profile in randomized study in non Mediterannean West Balkan country written by
Gordana Petrović-Oggiano1, Jasmina Debeljak-Martačić1, Slavica Ranković1, Biljana Pokimica1Alma Mirić1, Maria Glibetić1, Tamara Popović 1,
1. We tried to answer you in a best possible way in the text.Thanks to Your suggestion, we emphasised in abstract, materials and methods and results that our study was divided into substudies. The disscussion also follow this order mostly. We are thankful to you because now the text, aims and all parts of manustrips is much more understandable. This structure you suggested was more than necessary.
2. Walnuts as a source of n-3 fatty acids , mostly (ALA) are used in Serbian traditional meals or cakes, while fish intake is very low, as was estimated by 24 hour dietary recall. We thought that intake of walnuts could imrove n-3 intake and in our interventional short term study our results confirm it. Specially, statistical significance in EPA percentage more than in DHA was present.
3.Our study was randomized and 352 subjects answered to 24 hour dietary recall, for 110 of them (that we randomly selected) we analysed fatty acids plasma lipids profiles, while in 18 (also randomly chosen) we performed intervetion substudy, and they ate 56 g of walnuts daily for 4 weeks. Subjects were representative, we put in this version of article demographic table.
4. Regarding Ethical Comeettee, we had approvement and it was written in the line 121-122:Ethics Committee Approval) Clinical Hospital Center Zemun, Belgrade, Serbia, (No: 2125, 2013). That was enough for starting the study then. That approvment goes with explanation about not eating fish during 4 weeks of interventional study. It was written in the text and approved.
5. We tried to correct discussion and conclusion as you suggested, hoping that now it will fit more the topic. We included demographic table in study results.We put in disscussion the part of pathway transforming ALA into EPA to DHA, as you suggested.
The novelty of our manuscript was the fact that percentage of ALA transforming into EPA from walnuts in our interventional study was statistically significant. That significance is much more pronounced in our study comparing literature that we read and used for preparing the experiment.
6.At the end, as you said, at last but not at least, we changed a title of manuscript hoping that it will be suitable for the Special issue NUTS in Nutrients Journal.
At the end I am very thankful to you for your suggestions hoping that now when the structure of the text is more clear, the article could have bettter impact and fit into special issue of the very respectable Journal such as Nutrients.
Sincerely,
Dr sc.med.Gordana Petrovic Oggiano
Institute for medical research University of Belgrade
Laboratory for food and metabolism
Reviewer 2 Report
The manuscript contains numerous grammatical errors and could benefit from extensive grammatical editing. There are also methodological flaws. 18 subjects participated in an intervention with walnuts but its not clear how these 18 subjects were selected. Lack of randomization also makes this manuscript inferior to other publications that have looked at the same. Tables and figures are not well done. Table titles should be self-explanatory and include appropriate units There should have been a demographic table to explain who the subjects are. You failed to discuss an important point, that ALA has to be converted to EPA and DHA by enzymes in the body, and this process can be quite inefficient. There should have been a paragraph dedicated to this. Intake does not always translate to status because of this very reason. Fish intake in this population is somewhat decent. This in itself can affect ALA conversion in the body. The discussion could be enriched more. The manuscript lacks novelty
Author Response
Answers to Reviewer 2
Dear Reviewer ,
We are grateful for your useful and very constructive suggestions for our article named:
Effect of walnuts consumption on improvement of n-3 fatty acid profile in randomized study in non Mediterannean West Balkan country written by
Gordana Petrović-Oggiano1, Jasmina Debeljak-Martačić1, Slavica Ranković1, Biljana Pokimica1Alma Mirić1, Maria Glibetić1, Tamara Popović 1,
We tried to answer you in a best possible way in the text.
We explain in the text that study was randomised and that from 352 subjects (which did 24h recall) randomly 110 subjects were selected for experimental determination of plasma fatty acids profiles to present a lack of n-3 PUFA in Serbian population and randomly 18 were included in intervention study (money limitation- it was a pilot study) 56 g of walnuts daily for 4 weeks. We included demographic table as you suggest thinking that was very useful for the parameters we already had in study results, this version contain it. We put in discussion the part of pathway transforming ALA into EPA to DHA as you asked and explain this with desaturases and elongases activation. The novelty of our manuscript was the percentage statistically significant of ALA transforming into EPA from walnuts in our intervention study. That significance is much more highlighted in our study comparing literature that we read and used for preparing the experiment.
Hoping that our answer satisfies you and that new improved version is ameliorate with your suggestions of course, the special issue dedicated to nuts could this article read once more and decide. Without Your help it would not be the same and possible.
At the end I am very thankful to you for your suggestions hoping that now when the structure of the text is more clear, the article could have better impact and fit into special issue of the very respectable Journal such as Nutrients.
Sincerely,
Dr sc.med. Gordana Petrovic Oggiano
Institute for medical research University of Belgrade
Laboratory for nutrition and metabolism
Reviewer 3 Report
This manuscript covers several research questions and provides information on whether low intake of omega-3-fatty acids in form of fish can be balanced by consumption of walnuts (in Serbia). The manuscript covers several sub-studies: In the first sub-study, the consumption of fish and omega-3-fatty-acids in Serbia was evaluated by 24-hour recalls. The second sub study evaluates the composition of Serbian walnuts, the third sub-study evaluates fatty-acid-levels in healthy Serbians and the fourth sub-study evaluates the effect of walnut consumption on the fatty-acid profile in plasma.
Although those 4 aims are stated at the end of the introduction, in the methods and results section everything is presented in a rather confusing way. I suggest to structure the methods, the results and partially also the discussion according to the 4 aims (4 sub-studies). It may also be prudent to start with the data relating to the composition of walnuts and then go into the human studies (dietary recalls, plasma levels, intervention). This would make the study easier to understand for the reader.
In addition, several topics should be addressed:
How were the subjects selected? This refers to the subjects selected for dietary recall, plasma measurements and intervention. How did you make sure that these subjects are representative for the Serbian population? It seems that consent was only obtained for the sub-study where blood samples were analyzed – is that true? Please clarify. This reviewer has difficulties to understand how you prepared the walnut samples (line 69-73). Please clarify. The tables comparing the composition of walnuts from different European countries should move into the discussion section, as you did not measure the composition in walnuts from different countries.
Author Response
Answers to Reviewer 3
Dear Reviewer,
We are grateful for your useful and very constructive suggestions for our article named:
Effect of walnuts consumption on improvement of n-3 fatty acid profile in randomized study in non Mediterannean West Balkan country written by
Gordana Petrović-Oggiano1, Jasmina Debeljak-Martačić1, Slavica Ranković1, Biljana Pokimica1Alma Mirić1, Maria Glibetić1, Tamara Popović 1,
We tried to answer you in a best possible way in the text.
1.First of all We explain that our study had substudies as you already mention. Also we emphasized it in abstract, materials and methods and results. The discussion also follow this order mostly. We are thankful to You because now the text, aims and all parts of the manuscript is much more understandable. This structure you suggested was more than necessary.
2.Walnuts as a source of n-3 fatty acids , mostly (ALA) are used in Serbian traditional meals or cakes, while fish intake is very low, assessed by 24 hour dietary recall. We thought that intake of walnuts could improve n-3 intake and in our intervention short term study our results confirmed it. Specially statistical significance in EPA percentage more than DHA was present.
3. Our study was randomized and 352 subjects responded to 24 h recall, in 110 of them randomly we analysed fatty acids plasma lipids profiles while in 18 (randomly chosen) we performed intervetion substudy, and they ate 56 g of walnuts daily for 4 weeks. Subjects were representative, we put in this version of article demographic table (because we had data).
4. Preparing of walnuts sample for analysis was written in the section Material and Methods, now we hope that it is improved, thanks to Your advice.
5. We moved the Table with comparison of walnuts composition from different countries in the discussion section as you suggested.
At the end I am very thankful to You for Your suggestions hoping that now, when the structure of the text is more clear, the article could fit into special issue of the very respectable Journal such as Nutrients.
Sincerely,
Dr sc.med. Gordana Petrovic Oggiano
Institute for medical research University of Belgrade
Laboratory for food and metabolism
Round 2
Reviewer 1 Report
Overall comments
-Because of the extent of the improvements needed to ensure enough quality of the scientific writing of the Methods section, I would like to -once again- ask the authors to review the manuscript using an appropriate checklist, and now also append it for review. I did my best effort to point out (please see below, subheading “methods”) the most important issues to review. However, pointing out the entire list of descriptive issues that are missing is not doable, therefore I hereby insist on the need of using an internationally accepted checklist to reach scientific quality report to allow assessment of the study and then be able to evaluate its potential recommendation for publication in Nutrients.
-Should the reader understand that when the authors refer to “waltnuts” they always refer to Juglans regia L? This is relevant for a practical purpose, so that further studies could appropriately attempt to replicate the results of the intervention of this study. Please explicitly indicate which walnuts were analysed for nutrients and bioactive components in the first sub-study, and which walnuts were used for the intervention.
-the manuscript is much readable now, however, thorough and extensive english revision is still necessary and I highly recommend the use of english editing services.
Title
-it needs english correction
Abstract
-now the authors clearly describe 4 objectives. 1. Nutritional composition analyses of serbian walnuts. 2. Analyses of consumption of marine-derived n-3 PUFA in serbia. 3. Analyses of fatty acids profile in Serbian people. 4. The effects of a dietary intervention on fatty acids profile. Please, likewise, shortly describe results for everyone of these objectives in the abstract and in the Conclusion section of the manuscript.
-I do not consider it correct to say that the authors evaluated the consumption of fish and n-3 fatty acids in Serbia. Please mind the limitations of the study population.
Introduction
-it now reads very well. My only comment is that “the gap” (line 52), is not explicitly stated. The authors describe the problem about the shift on n-6/n-3 PUFA due to Western diet, but that does not necessarily imply a “gap”. It would add if the authors provide a sentence with appropriate bibliography to actually indicate and support “the gap” to which they refer, so that it is not subjective to interpretations.
Methods
-please do not describe the study population in the methods section but in the results section. Instead, in the methods section please describe recruitment approach.
-how many patients were approached for recruitment
-how did the authors actually evaluated that the participants were “healthy”? please describe in detail.
-Is it possible to say that a population with a BMI of app 29 is indeed healthy?
-how many patients rejected participation.
-how did the authors determine to select 110 study subjects. Please provide sample size calculation.
-how did the authors determine to select 18 study subjects for the intervention. Please provide sample size calculation.
-which method was used to randomly select these sub-samples of patients?
-did all the patients accepted to participate in the intervention a priori (before being actually selected to participate in this specific section of the study), or were the patients approached for participation in the intervention sub-study at a later stage? Please mention this in the manuscript.
-how were the participants instructed to consume 56 g/d of walnuts?
-Did all the patients finished the intervention?
-How did the authors evaluated dropouts thorughout?
-please do not mix the description of statistical analyses in between the description of the performance of the study itself.
Discussion
-please start by writing a paragraph that summarizes all the findings. After that, please start the discussion.
-please amend the writing in such a way that every paragraph starts with a title sentence, easily indicating the topic that will be discussed in the paragraph. And then please finish every paragraph with a sentence that provides the reader with the conclusion that the authors propose regarding the corresponding idea discussed in the paragraph.
Other comments:
-the authors sometimes refer to “other three nuts”. I could not find, however, explicit information stating which nuts they are referring to.
-please rewrite line 169.
-please suppor the statements in line 174-177 with appropriate bibligraphy.
Author Response
Dear Reviewer 1.,
We send to You our Responses regarding Your Valuable and helpful Comments.You can see below.
Overall comments
-Because of the extent of the improvements needed to ensure enough quality of the scientific writing of the Methods section, I would like to -once again- ask the authors to review the manuscript using an appropriate checklist, and now also append it for review. I did my best effort to point out (please see below, subheading “methods”) the most important issues to review. However, pointing out the entire list of descriptive issues that are missing is not doable, therefore I hereby insist on the need of using an internationally accepted checklist to reach scientific quality report to allow assessment of the study and then be able to evaluate its potential recommendation for publication in Nutrients.
The Methods section of the Manuscript has been changed according to comments provided below. We provide answers to every single question. In addition, we have used the CONSORT flow chart and checklist (please see the attachment) to make sure that, within our description of the study protocol, we have covered all important aspects related to our study design and its implementation.
-Should the reader understand that when the authors refer to “walnuts” they always refer to Juglans regia L? This is relevant for a practical purpose, so that further studies could appropriately attempt to replicate the results of the intervention of this study. Please explicitly indicate which walnuts were analysed for nutrients and bioactive components in the first sub-study, and which walnuts were used for the intervention.
In this study one particular type of walnuts has been used. It is Juglans regia L. We made this clear in the Manuscript, line 32 and line 60. The same type of walnuts has been used both in the first sub-study and in the intervention trial.
-the manuscript is much readable now, however, thorough and extensive english revision is still necessary and I highly recommend the use of english editing services.
Thank you for this comment. The Manuscript has been checked by a native English speaker and grammatical and spelling errors have been rectified. In addition, English language expressions are corrected as needed.
Title
-it needs english correction
The title has been changed/rewritten:
“The effect of walnut consumption on n-3 fatty acid profile of healthy people living in a non-Mediterranean West Balkan country, a small scale randomized study”
Abstract
-now the authors clearly describe 4 objectives. 1. Nutritional composition analyses of serbian walnuts. 2. Analyses of consumption of marine-derived n-3 PUFA in serbia. 3. Analyses of fatty acids profile in Serbian people. 4. The effects of a dietary intervention on fatty acids profile. Please, likewise, shortly describe results for everyone of these objectives in the abstract and in the Conclusion section of the manuscript.
The results for each of the four objectives have been provided both in the abstract (lines 18, 20-21, 24-26) and in the conclusion section of the manuscript (lines 307-318).
-I do not consider it correct to say that the authors evaluated the consumption of fish and n-3 fatty acids in Serbia. Please mind the limitations of the study population.
Thank you for making this comment. The sentence has been rewritten to clearly demonstrate the data obtained through this study.
The second sub-study assessed the consumption of walnuts (Juglans regia L.) and total n-3-fatty acid intake in apparently healthy Serbian residents, using 24-hour dietary recalls (n= 352) (lines 18-20)
Introduction
-it now reads very well. My only comment is that “the gap” (line 52), is not explicitly stated. The authors describe the problem about the shift on n-6/n-3 PUFA due to Western diet, but that does not necessarily imply a “gap”. It would add if the authors provide a sentence with appropriate bibliography to actually indicate and support “the gap” to which they refer, so that it is not subjective to interpretations.
This section of the manuscript has been rewritten and English language expressions are corrected as needed. The term ‘gap’ has been replaced with a ‘low dietary intake of n-3 FAs’.
Methods
-please do not describe the study population in the methods section but in the results section. Instead, in the methods section please describe recruitment approach.
All of the sentences that described the study population in the Methods section have been removed to the Results section of the manuscript.
-how many patients were approached for recruitment
The study participants were apparently healthy volunteers recruited through primary health care facilities where flyers about participation in the study were distributed. ‘A total of 400 subjects were approached for the recruitment via the Institute for medical research and 352 were involved in the dietary data analysis. Reasons for drop out are mentioned in new Figure 1.
-how did the authors actually evaluated that the participants were “healthy”? please describe in detail.
Apparently healthy’ was defined as the absence of any clinical signs of acute condition or chronic disease and without the need for a medical treatment.
-Is it possible to say that a population with a BMI of app 29 is indeed healthy?
You are right, thank you for this comment. That is why we added “apparently”.
-how many patients rejected participation.
A total of 11 rejected, as we described in the new flow diagram.
-how did the authors determine to select 110 study subjects. Please provide sample size calculation.
For the analysis of fatty acids in plasma phospholipids 30% of initially enrolled subjects were randomly selected. Blood samples were obtained from 114 participants. We performed analysis of 110 participants, because 4 samples were damaged during analysis.
-how did the authors determine to select 18 study subjects for the intervention. Please provide sample size calculation. which method was used to randomly select these sub-samples of patients?
As our intervention study was a small scale (pilot study) 10%-15% of study participants was intended to be included. 10-15% of 114 is (11-17 participants). We have to mention here that due to restricted resources we were not able to include more participants within this interventional sub-study.
-did all the patients accepted to participate in the intervention a priori (before being actually selected to participate in this specific section of the study), or were the patients approached for participation in the intervention sub-study at a later stage? Please mention this in the manuscript.
At the later stage, as mentioned in line 128.
-how were the participants instructed to consume 56 g/d of walnuts?
Based on literature data, mentioned in text, line 133.
-Did all the patients finished the intervention?
Yes.
-How did the authors evaluated dropouts throughout?
There was no dropouts. We were in contact via telephone with all participants during the intervention period.
-please do not mix the description of statistical analyses in between the description of the performance of the study itself.
Thank you for this comment. We hope that now we improved the text.
Discussion
-please start by writing a paragraph that summarizes all the findings. After that, please start the discussion.
A short summary of major findings obtained in this study has been provided at the beginning of the Discussion section, lines 226-229.
-please amend the writing in such a way that every paragraph starts with a title sentence, easily indicating the topic that will be discussed in the paragraph. And then please finish every paragraph with a sentence that provides the reader with the conclusion that the authors propose regarding the corresponding idea discussed in the paragraph.
Thank you for this comment. We tried to follow Your suggestions.
Other comments:
-the authors sometimes refer to “other three nuts”. I could not find, however, explicit information stating which nuts they are referring to.
The expression ‘other three nuts’ has been defined to make clear that we are referring to almonds, .….
We have also explained this in the paper, lines 240-241, 274-275.
-please rewrite line 169.
The sentence (lines 169-171) has been rewritten as suggested.
-please suppor the statements in line 174-177 with appropriate bibligraphy.
The statement made (lines 174-177) has been cited appropriately. Thank you.

Reviewer 2 Report
The manuscript has improved, and reads much better than the first draft.
Author Response
Dear Reviewer 2.,
Thank You very, very much for Your Valuable Comments.
Reviewer 3 Report
Thank you for the revised version of the manuscript and for incorporating the suggested changes. I believe that the manuscript has improved considerably.
I suggest to use subheadings in the results section to better structure the section the manuscript must be corrected by a native English speaker! There are numerous errors (examples: line 125, "they" missing; line 170 "was" should be "is"; line 177, "s" redundant; line 182, "a" redundant; lines 276-280, hard to understand).
Author Response
Thank you for this comment. The Manuscript has been checked by a native English language speaker and grammatical and spelling errors have been rectified. In addition, the English language expressions are corrected as needed.